# Cohort profile: Life-course experiences and pregnancy (LEAP)–A retrospective pregnancy cohort for life-course reproductive health research

Susan M. Mason[ID][1]*, Kriszta Farkas[1], Jessica K. Friedman[2], Anne Gerlach[1], Sydney T. Johnson[1], Rebecca Emery Tavernier[3,4], Lisa M. Bodnar[5], Dianne Neumark-Sztainer[1]

1 Division of Epidemiology and Community Health, University of Minnesota School of Public Health, Minneapolis, MN, United States of America, 2 Center for Care Delivery and Outcomes Research (CCDOR), Minneapolis Veterans Healthcare System, Minneapolis, MN, United States of America, 3 Weitzman Institute, Middletown, CT, United States of America, 4 Department of Family and Biobehavioral Health, University of Minnesota Medical School, Duluth Campus, Duluth, MN, United States of America, 5 Department of Epidemiology, University of Pittsburgh School of Public Health, Pittsburgh, PA, United States of America

* smmason@umn.edu

**Data Availability Statement:** Because the data contain sensitive information, including Protected Health Information, access is restricted per

## Abstract

### Background

Life course factors may be associated with pre-pregnancy body mass index and gestational weight gain; however, collecting information on pre-pregnancy exposures and pregnancy health in the same cohort is challenging.

### Objectives

The Life-course Experiences And Pregnancy (LEAP) study aims to identify adolescent and young adult risk factors for pre-pregnancy weight and gestational weight gain (GWG). We built upon an existing cohort study to overcome challenges inherent to studying life course determinants of pregnancy health.

### Population

Participants in an ongoing prospective cohort study of weight-related health who identified as women.

### Design

Retrospective cohort study.

### Methods

In 2019–2020, 1,252 women participating since adolescence in a cohort study of weight-related health were invited to complete an online reproductive history survey. Participants who reported a live birth were invited to release their prenatal, delivery, and postpartum medical records for validation of survey reports. Descriptive analyses were conducted to

University of Minnesota ethics committee guidelines. Data can be accessed via OpenICPSR at http://doi.org/10.3886/E193505V1. Once on the project page, click on the "Request Information" button on the right. This will take you to the data request form. More information on accessing restricted data from openICPSR can be found at https://www.openicpsr.org/openicpsr/accessRD.

**Funding:** This work was funded by R01HD090053 from the Eunice Kennedy Shriver National Institute of Child Health and Human Development (PI: SM; https://www.nichd.nih.gov/). The funders had no role in study design, data collection and analysis, decision to publish, or preparation of the manuscript.

**Competing interests:** The authors have declared that no competing interests exist.

assess the characteristics of the overall cohort and the medical record validation subsample, and to describe adolescent and young adult characteristics of those with high (>80th percentile), moderate (20th-80th percentile), and low (<20th percentile) GWG z-score for gestational age and pre-pregnancy weight status.

## Preliminary results

Nine hundred seventy-seven women (78%) completed the LEAP survey and 656 reported a live birth. Of these, 379 (58%) agreed to release medical records, and 250 records were abstracted (66% of the 379). Of the 977 survey respondents 769 (79%) reported attempting a pregnancy, and 656 (67%) reported at least one live birth. The validation subsample was similar to the overall cohort. Women with a high GWG had a higher adolescent BMI percentile and prevalence of unhealthy weight control behaviors than those with moderate or low GWG.

## Conclusions

LEAP offers a valuable resource for identifying life course factors that may influence the health of pregnant people and their offspring.

## Introduction

Body weight during the perinatal period, including pre-pregnancy weight and gestational weight gain (GWG), has important influences on subsequent maternal and infant health. Individuals who enter pregnancy at a higher body mass index (BMI) are at greater risk for pregnancy and delivery complications, such as gestational diabetes, pre-eclampsia, and cesarean delivery [1, 2]. Similarly, high GWG may increase risk for certain pregnancy complications [3] and for later obesity and cardiovascular dysregulation for the pregnant person [4, 5]; infants born after pregnancies characterized by high GWG tend to have higher adiposity in childhood [6]. Identifying who is most at risk for weight-related problems during the perinatal period, and what factors influence that risk, remains an important challenge.

Although many risk factors for high pre-pregnancy BMI have been identified, very few factors have been identified that consistently predict high GWG. Many risk factors for higher pre-pregnancy BMI (e.g., older age) are either not associated with excess GWG or show an inverse association [7, 8]. One of the few strong and consistent predictors of high GWG (above medical recommendations [7]) is pre-pregnancy overweight or obesity [9, 10]. However, this is largely an artifact of the GWG recommendations themselves, which are set lower for people with higher pre-pregnancy body weights (e.g., for those defined as having obesity prior to pregnancy, the IOM recommends GWG of 11–20 pounds, versus 25–35 pounds in those defined as having normal pre-pregnancy weight [7]). Other risk factors for high GWG remain largely unidentified.

There is increasing evidence that risks for many health outcomes are shaped by factors that accumulate and interact over the life course [11, 12]. For example, studies indicate that early life exposures, such as adverse childhood experiences and childhood nutrition, may set the stage for future cardiovascular risk [13–16]. The emergence of life course theory and evidence highlighting the sustained impacts of early life exposures on the development of disease in later life has been an important paradigm shift in understanding disease etiology.

As with other health outcomes, health during pregnancy is likely shaped by exposures across the life course. For example, dieting, unhealthy weight control behaviors, and binge eating in adolescence are associated with accelerated weight gain trajectories and higher body weight in adulthood [17, 18]. These associations may extend into pregnancy, yet almost nothing is known about how a history of adolescent disordered eating or unhealthy weight control behaviors may influence weight behaviors and weight gain during pregnancy. Likewise, a history of childhood maltreatment, which is associated with disordered eating and obesity in non-pregnant adults [19–25], could potentially be a risk factor for disordered eating [26] and excessive weight gain in pregnancy [27–29].

Despite the plausibility of links between adolescent risk factors and pregnancy weight, there has been limited empirical research on these types of life course associations. A key barrier to such work is the difficulty of collecting information on both early life exposures and pregnancy-related health in a single cohort. Most studies of pregnancy outcomes rely on recruiting people during pregnancy and following them to delivery, limiting capacity to accurately assess earlier life risk factors that may be difficult to recall (e.g., adolescent dieting). On the other hand, studies with data from earlier life face major feasibility challenges in collecting prospective pregnancy data; it is estimated that only 7% of women of reproductive age will give birth in any given year [30], and almost half of those births will be unintended [31], making prospective collection of pregnancy data in existing cohorts extremely difficult. Thus, to date, studies examining GWG have not had information on the broad range of childhood and adolescent exposures and weight behaviors that would allow comprehensive investigation of life course risk factors for excessive GWG.

One potential solution to these challenges is the collection of retrospective data on pregnancies of participants in studies with previously-collected data from the earlier life course. In this paper we describe building a retrospective pregnancy cohort, the Life-course Experiences And Pregnancy Study (LEAP) study, embedded within an ongoing longitudinal study of weight-related health and behavior from adolescence to young adulthood, Project EAT (Eating and Activity in Teens and Young Adults). Project EAT provides a rich set of prospectively-collected data on weight-related risk factors, such as binge eating and unhealthy weight control behaviors, across the life course. Once recruited into LEAP, participants were asked to recall key characteristics of their pregnancies, including pre-pregnancy body weight and GWG. To address potential recall bias, we validated survey reports of pre-pregnancy weight, GWG, and pregnancy complications (e.g., gestational diabetes) against medical record data in a validation subsample. One prior study has similarly linked pregnancy data to earlier-life information: the National Longitudinal Survey of Youth (NLSY) asked participants to retrospectively recall their pre-pregnancy weight and GWG. However, the NLSY has limited data on weight-related risk factors over the life course, and pregnancy measures in the NLSY have not been validated. Thus, the LEAP cohort is unique in linking adolescent weight-related risk factors with validated weight-related pregnancy measures.

The LEAP cohort was designed to support research into life course determinants of pregnancy weight-related health, and to fill a major gap in existing data linking adolescent weight-related risk factors to validated pregnancy outcomes. This linkage across the life course will allow us to assess whether certain environmental, psychological, or behavioral factors in adolescence predict weight measures in pregnancy, informing an understanding of the life course etiology of pregnancy weight-related health. The specific focus of LEAP is to identify the extent to which childhood maltreatment and its sequelae, including adolescent and young adult disordered eating for depression, are linked to high pre-pregnancy BMI and excessive GWG. The conceptual model guiding the LEAP study is shown in Fig 1, which illustrates the theory that child maltreatment victimization reduces the capacity for *affect regulation* (the ability to cope

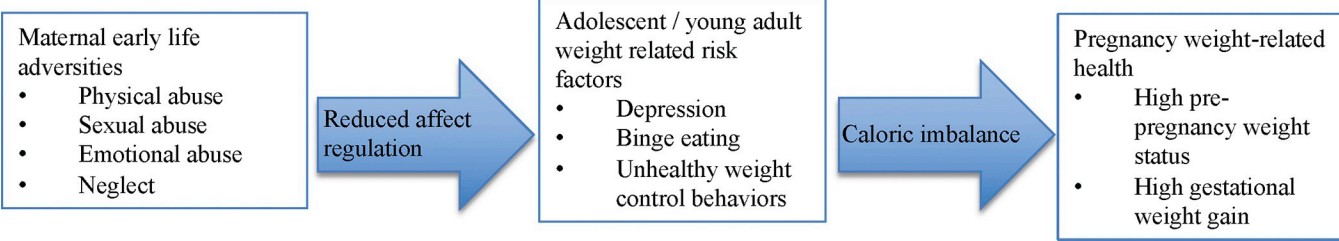

**Fig 1. Conceptual model guiding the LEAP study. Boxes present measured variables, and arrows are hypothesized pathways.**

with distress), leading to obesity-promoting behaviors that are soothing or numbing, such as overeating of highly palatable foods that trigger dopaminergic reward responses [32–37]. Associations between child maltreatment, disordered eating, and body weight have been repeatedly demonstrated outside of pregnancy [19–25]. To disseminate information about this dataset, the goals of the current paper are to describe: (1) the study design, recruitment, and demographic and pregnancy characteristics of the overall LEAP cohort and the medical record validation subsample; and (2) key adolescent characteristics of those with standardized GWG that was low (<20th percentile), moderate (20th– 80th percentile), and high (>80th percentile) given their pre-pregnancy weight status and gestation length.

## Materials and methods

### Cohort description

**Study design and sample.** *LEAP Cohort*. LEAP is a retrospective cohort study of reproductive and perinatal health among 977 women participating since adolescence in an ongoing longitudinal study of eating, activity, and weight-related health (Project EAT). For Project EAT, male and female adolescents aged 11–18 were recruited during the 1998–1999 school year from 31 public middle and high schools in the greater Minneapolis-St. Paul area of Minnesota. Participants were subsequently surveyed every five years, with the most recent survey completed in 2014–2015 when participants were aged 25–36. As described in detail elsewhere [38], surveys have collected data on a wide range of measures including depressive symptoms, unhealthy weight control behaviors, binge eating, home food availability, physical activity, and body weight.

The LEAP cohort was enrolled between July 2019 and July 2020. Participants in Project EAT who identified as women were recruited to complete the LEAP survey if they had responded to at least two of the three most recent Project EAT surveys and resided in the United States. The survey was administered online via REDCap. Participants were first taken through an electronic informed consent process in which they were provided with information about the study and asked to click "I agree to participate in this research" to indicate their consent. After consenting, participants were taken to the survey. The LEAP survey asked participants to report on their reproductive and pregnancy histories, including how many pregnancies and live births they have had. Invitations to participate were sent by email and/or mail, and women were offered a $50 incentive to complete the survey. To enhance participant response, email and mail reminders were sent to nonrespondents, one of which included a $2 bill to increase attention to the mailing. In the final months of recruitment an additional $25 gift card incentive was offered for participation.

*Validation substudy*. All participants who reported at least one live birth on the LEAP survey were invited to sign HIPAA Authorization and medical records release forms authorizing

the study team to request their prenatal, delivery, and postpartum medical records for each of their pregnancies. Return of the signed forms served as informed consent for the validation substudy. Substudy participants were offered an additional $50 gift card incentive. All study protocols were approved by the University of Minnesota Institutional Review Board Human Subjects Committee.

**Data collection.** *LEAP survey*. The 2019–2020 LEAP survey was administered online using REDCap (Research Electronic Data Capture) [39]. Most items included in the LEAP survey were adapted from validated instruments or items on existing surveys (e.g., Project EAT, the Nurses' Health Study, Pregnancy Risk Assessment and Monitoring Survey); in cases where appropriate existing measures could not be located, the study team developed items specifically for LEAP. The LEAP survey was pilot tested in a convenience sample of 20 mothers recruited on social media, and feedback was used to refine the survey before fielding it in the LEAP cohort. Test-retest reliability of LEAP survey measures was assessed in a random sample of 141 LEAP survey respondents who completed the survey a second time within a two-week period. The LEAP questionnaire asked participants about their reproductive histories, including whether they had ever attempted to get pregnant, whether they had been pregnant, and how many of their pregnancies resulted in a live birth. Women who reported having at least one live birth were asked, for each birth reported: their age when they gave birth; the gestation length of the pregnancy; any diagnosed pregnancy complications; and their best estimates of what they weighed just before the pregnancy and how much weight they gained during the pregnancy. In addition, for their first live birth only, women were asked about a range of psychosocial and behavioral characteristics during the pregnancy, including their perceived ability to live on their income, depressive symptoms, binge eating, and unhealthy weight control behaviors, among others. Women who had never been pregnant were asked if they had ever attempted to get pregnant and, if so, if they had received a diagnosis of infertility and/or sought fertility treatment. A subset of measures assessed on the survey, with test-retest values, is presented in Table 1. Survey questions are provided in the S1 Appendix in S1 File.

*Medical record abstraction*. Self-reports of key pregnancy characteristics for each woman's live births were validated against medical record data in a validation substudy. The validation subsample was comprised of women who returned HIPAA Authorization and medical record release forms allowing the study team to obtain their prenatal, delivery, and postpartum records for each of their live births (Fig 2). After medical records were obtained, trained study staff abstracted data on pre-pregnancy weight, measured weights at each prenatal visit, delivery weight, gestational age at delivery, and diagnoses of pregnancy complications and entered these data into REDCap. Prior to starting data abstraction and entry, all staff were required to test to a minimum inter-rater reliability (Cohen's Kappa) value of 0.80 on one medical record for fidelity; reliability across abstractors ranged from 0.84 to 0.97. Additionally, to ensure reliability across data abstractors, double data entry was performed for 61 (24%) randomly sampled medical records, which showed high inter-rater reliability (Cohen's Kappa: 0.97).

*Project EAT data*. Four Project EAT surveys had been conducted prior to the initiation of the LEAP study. At Project EAT baseline (EAT-I, 1998–1999; ages 11–18), participants were asked to self-report key sociodemographic characteristics including their race and ethnicity, their parents' educational attainment, and whether they received free or reduced-price school lunch. Participants were asked to self-report their height and weight and had their height and weight measured by study staff for validation. At each EAT wave, participants were asked to self-report several weight-related risk factors and characteristics including their body weight (and, prior to age 18, their height); binge eating in the past 12 months, and whether they had experienced loss of control while binge eating; unhealthy weight control behaviors (e.g., fasting, purging, using diet pills); body dissatisfaction; and depressive symptoms. At EAT-IV

**Table 1. Subset of reproductive and pregnancy history variables assessed on the LEAP survey.**

| Variable | Question and response options | Test-retest reliability | Derived variables |
|---|---|---|---|
| Reproductive history | How many total pregnancies have you had, including pregnancies resulting in live births, stillbirths, miscarriages, or termination? (Do not count a current pregnancy) [0–15] | r = 0.99 | Gravid (1+ pregnancies) vs. Non-gravid (0 pregnancies) |
| Live birth | How many of your pregnancies have resulted in a live birth? If a multiple pregnancy (resulting in two or more babies), count as ONE birth. [0–10] | r = 0.79 | Parous (1+ live births) vs. Non-parous (0 live births) |
| Multiple birth | Was your [first] live birth a multiple birth (giving birth to or delivering two or more babies)? [Yes; No] | κ = 1.0 | Singletons vs. Multiple |
| Infant birth weight, first birth | Approximately how much did [baby's name] weigh at birth? (For reference, the average birth weight for babies is around 7.5 pounds, although between 5.5 and 10 pounds is considered normal. There are 16 ounces in 1 pound.) [Open-ended] | r = 0.99 | Continuous in kgs; sex-specific weight-for-gestational-age z-score [40] |
| Gestational age at birth, first birth | How many weeks pregnant were you when this baby was/these babies were born (i.e. gestational age at birth)? (For reference, 40 weeks is an average gestation for a singleton, and 38 weeks for twins) [Open-ended] | r = 0.98 | Continuous gestational age in weeks; Preterm (<37 weeks) / Term (≥37 weeks) |
| Infant sex, first birth | What was [baby's name]'s sex at birth [Male; Female; I don't know; Prefer not to answer] | κ = 0.98 | |
| Neonatal health complications, first birth | Did [baby's name] have any medical or health problems at birth? [Check all that apply: None; Fever/infection; Respiratory issues; Heart problems; Blood disorder; Prematurity; Required NICU or special care; Other] | κ ranged from 0.66 (other) to 1.00 (Fever/infection, blood disorder, prematurity) | |
| Pre-pregnancy weight, first birth | About how much did you weigh, in pounds, just before this pregnancy? [Open-ended] | r = 0.96 | Body mass index (weight in kgs / [height in meters]$^2$) using reported height from EAT-IV survey |
| GWG, first birth | About how much weight, in pounds, did you gain during this pregnancy? [Open-ended] | r = 0.94 | GWG z-scores computed based on gestational age at delivery and pre-pregnancy BMI category [41, 42] and categorized as high (>80th percentile), moderate (20th-80th percentile) and low (<20th percentile). |
| Pregnancy complications, first birth | During this pregnancy, were you ever told by a doctor, midwife, or other medical provider that you had any of the following conditions? [Check all that apply: Gestational diabetes; Pregnancy-related high blood pressure; Pre-eclampsia/toxemia; Hyperemesis gravidarum (excessive throwing up); Post-traumatic stress disorder, eating disorder or other psychiatric disorder (e.g. mood, anxiety); A condition that required bed rest; Other medical condition we should be aware of; None] | κ ranged from 0.38 (bed rest) to 1.0 (Diabetes, Pre-eclampsia, Psychiatric disorder) | |
| Cesarean delivery, first birth | Did you have a cesarean section (c-section) with this birth? [Yes; No] | κ = 1.0 | |

Abbreviations: EAT, Eating and Activity in Teens and Young Adults study; GWG, gestational weight gain; κ, Cohen's kappa coefficient; r, Pearson correlation.

Note: Where measure is specific to one birth, the test-retest reliability measures presented are for the first live birth.

(aged 25–36), participants were asked to retrospectively report experiences of childhood abuse and neglect, which have been found to be associated with adult body weight [23, 43, 44], using measures from the Childhood Trauma Questionnaire [45] and survey questions by Finkelhor et al. [46] Table 2 provides details on a selection of key Project EAT measures, including survey questions, test-retest values, and variables derived from each measure.

**Analysis variables.** The current paper uses the following variables from Project EAT and LEAP to describe the LEAP cohort. Pregnancy-related variables are for the pregnancy resulting in the first live birth. Details are provided in Tables 1 and 2.

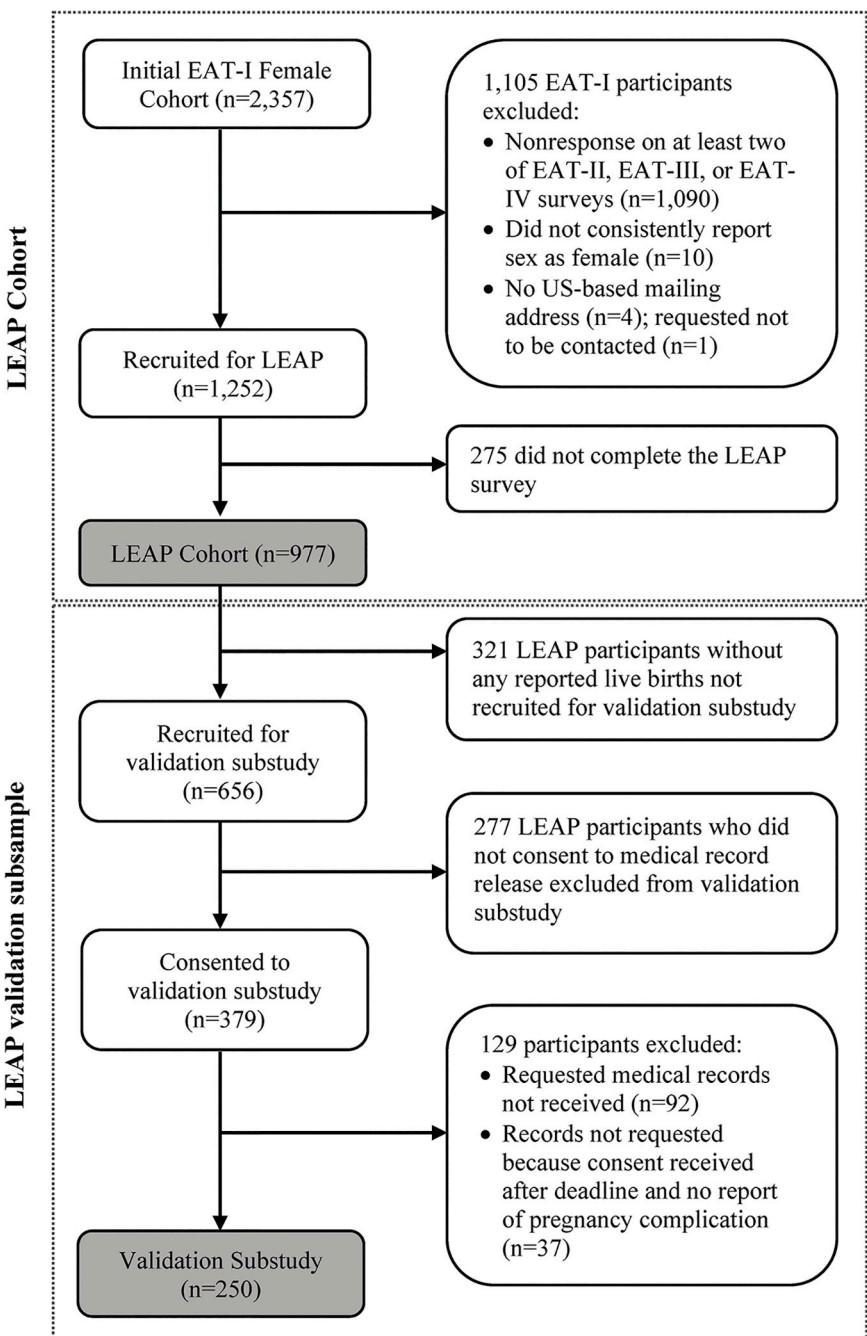

**Fig 2. LEAP cohort and validation subsample recruitment.**

*Pre-pregnancy BMI*. Pre-pregnancy BMI was self-reported on the LEAP survey in pounds and converted to kg/m$^2$ using self-reported height from the LEAP, EAT-IV, or EAT-III surveys. Pre-pregnancy BMI was examined as a continuous variable to describe the mean and SD in the sample and was categorized into under/normal weight (BMI < 25 kg/m$^2$), overweight (25 kg/m$^2$ ≥ BMI < 30 kg/m$^2$), and obesity (BMI ≥ 30 kg/m$^2$) for computation of GWG z-scores specific to pre-pregnancy weight status.

**Table 2. Subset of sociodemographic and weight-related variables on the Project EAT surveys.**

| Variable | Question and response options | Test-retest reliability | Derived variables |
|---|---|---|---|
| Race and ethnicity (EAT-I) | Do you think of yourself as. . . (You may choose more than one) [White; Black; Hispanic or Latino; Asian American; Hawaiian or Pacific Islander; American Indian or Native American] | κ = 0.70–0.83 [47] | Categorized as White, Black, Hispanic, Asian, and Other |
| Childhood Socioeconomic status (EAT-I) | How far in school did your mother go? (Indicate the highest level) [Did not finish high school; Finished high school or got GED; Did some college or training after high school; Finished college; Master's degree or PhD; I don't know] How far in school did your father go? (Indicate the highest level) [Did not finish high school; Finished high school or got GED; Did some college or training after high school; Finished college; Master's degree or PhD; I don't know] Do you qualify for free or low-cost lunch? [Yes; No; I don't know] Does your family get public assistance (welfare, food stamps, or other assistance)? [Yes; No; I don't know] | κ = .79 (mother) κ = 0.78 (father) κ = 0.79 (lunch) κ = 0.80 (public assistance) | Childhood socioeconomic status was based on the highest educational attainment of either parent. Missing and implausible values were imputed using information about public assistance and free and reduced price lunch [48] |
| Adolescent BMI percentile (EAT-I) | How tall are you? [open-ended] How much do you weigh? [open-ended] | r = 0.60 (height) r = 0.58 (weight) | BMI was calculated as weight in kg / [height in meters]$^2$. Sex-specific BMI-for-age percentiles were then calculated based on CDC standard growth charts [49, 50] |
| Binge eating (EAT-I) | In the past year, have you ever eaten so much food in a short period of time that you would be embarrassed if others saw you (binge-eating)? [Yes; No] If yes. . . During the times when you ate this way, did you feel you couldn't stop eating or control how much you were eating? [Yes; No] | K = 0.64 (large amount of food) κ = 0.23 (loss of control) | Binge eating with loss of control was defined as 'yes' when both items were positively endorsed, otherwise defined as 'no' |
| Unhealthy weight control behaviors (EAT-I) | Have you done any of the following things in order to lose weight or keep from gaining weight in the past year? (Yes/No) [Fasted; Ate very little food; Took diet pills; Made myself vomit; Used laxatives; Used diuretics (water pills); Used food substitute (powder / special drink); Skipped meals; Smoked more cigarettes] | κ ranged from 0.29 (laxatives) to 0.67 (skipped meals) | A dichotomous variable was created to identify those engaging in any behavior (yes/no for each method; po = 83%) [17] |
| Body dis-satisfaction (EAT-I) | Body dissatisfaction was assessed using a modified version of the Body Shape Satisfaction Scale.[51] Participants rated their satisfaction with 10 different body parts (height, weight, body shape, waist, hips, thighs, stomach, face, body build, shoulders). [5-point Likert scale: Very dissatisfied to Very satisfied] | r = 0.82 | Each item is scored from 1 (very dissatisfied) to 5 (very satisfied). The mean scores is taken and multiplied by 10, yielding a possible range of 10–50. High dissatisfaction was defined by a score of <30 |
| Depressive symptoms (EAT-I) | During the past 12 months, how often have you been bothered or troubled by. . . (Not at all, Somewhat, Very Much) [Feeling too tired to do things; Having trouble going to sleep or staying asleep; Feeling unhappy, sad, or depressed; Feeling hopeless about the future; Feeling nervous or tense; Worrying too much about things] | κ ranged from 0.27 (too tired) to 0.61 (unhappy/ sad/ depressed) | Each item is scored from 1 (not at all) to 3 (very much). The average of all scores is taken and multiplied by 10. A score of 23 or higher is considered indicative of clinically significant depressive symptoms [52] |

*(Continued)*

**Table 2.** (Continued)

| Variable | Question and response options | Test-retest reliability | Derived variables |
|---|---|---|---|
| Childhood abuse or neglect history (EAT-IV) | Before age 18 years, did someone in your family touch you in a sexual way or force you to touch them in a sexual way? [Never; Once; More than Once] Before age 18 years, did someone not in your family touch you in a sexual way or force you to touch them in a sexual way? [Never; Once; More than Once] When you were growing up, how often were the following statements true? [Never; Rarely; Sometimes; Often; Very often] -An adult in my family said hurtful or insulting things to me. -An adult in my family hit me so hard it left me with bruises or marks. -My family was a source of strength and support. | κ (familial sexual abuse) = 0.90 κ (non-familial sexual abuse) = 0.94 κ (physical abuse) = 0.69 κ (emotional abuse) = 0.63 κ (emotional neglect) = 0.59 | Sexual abuse items were derived from Finkelhor et al. [46] and physical and emotional abuse and emotional neglect items from the Childhood Trauma Questionnaire.[45] Responses were dichotomized following cut-points used in the ACE survey [53]. Both sexual abuse and physical abuse (hit so hard it left bruises) were dichotomized at any vs. none. Emotional abuse (family said hurtful things) was dichotomized as often or more vs. less. Emotional neglect (family was a source of strength and support) was dichotomized at rarely or less vs. more |
| Adult educational attainment (EAT-IV) | What is the highest level of education that you have completed? [Middle or junior high; Some high school; High school graduate or GED; Vocational, technical, or other certification program; Associate degree; Bachelor's degree; Graduate or professional degree (MS, MBA, MD, PhD); Other] | κ (>high school vs. ≤high school) = 0.77 | |

Abbreviations: BMI, body mass index; EAT, Eating and Activity in Teens and Young Adults study (number indicates survey wave: EAT-I is baseline at age 11–18 and EAT-IV is wave four at age 25–36); κ, Cohen's kappa coefficient; r, Pearson correlation.

*GWG.* Total GWG in pounds was self-reported on the LEAP survey. We standardized total GWG using z-score charts based on gestational age at delivery and pre-pregnancy BMI category [41, 42]. GWG z-scores provide a measure of GWG that is independent of gestational duration [54]. We categorized GWG z-scores as high (>80th percentile), moderate (20th-80th percentile), and low (<20th percentile).

*Sociodemographics.* Race (White, Black, Hispanic, Asian, and Other) was self-reported on EAT-I. Childhood socioeconomic status was based on participants' reports of their parents' educational attainment on EAT-I, with missing or implausible values imputed based on other markers of financial well-being (e.g., being on public benefits). Adult educational attainment (high school education or less, some college or vocational training, and Bachelor's degree or higher) was measured on the EAT-IV survey.

*Childhood risk factors and health indicators.* Adolescent BMI percentile based on CDC growth charts was derived from height and weight measured at the time of the EAT-I survey. Binge eating and unhealthy weight control behaviors in the past 12 months, body dissatisfaction, and depressive symptoms were taken from the EAT-I survey. A history of child maltreatment was retrospectively reported on EAT-IV when participants were adults (age 25–36).

*Reproductive and perinatal characteristics* were all measured on the LEAP survey and included: whether the participant had ever attempted a pregnancy, and whether they had had a live birth. Among those reporting live births, age at the birth, the gestation length of the pregnancy, whether the birth was a multiple, pre-pregnancy weight, and total GWG were also examined.

**Preliminary analyses.** Preliminary analyses were descriptive and designed give an overview of the key characteristics of the LEAP cohort overall and the medical record validation subsample. In addition, we present characteristics of those with low, moderate, and high GWG z-scores.

First, we calculated the distribution (number and percent for categorical and dichotomous variables, means and SDs for continuous variables) of women in the overall sample across the

sociodemographic, childhood and adolescent risk factor and health indicator, and reproductive perinatal variables described above. To allow for exploration of the extent to which the validation subsample differed from the overall cohort, we also present the distribution of the subsample across the same characteristics. Finally, because a primary outcome of interest in the LEAP study is GWG, we examine the distribution of these characteristics across categories of GWG.

We did no hypothesis testing, given growing consensus about the limited information it provides as well as the harms of its widespread use [55, 56], and instead focus on descriptive statistics and qualitative comparison across GWG groups.

## Results and discussion

### Findings to date

**Recruitment and response rate.** Of the 1,252 women invited, 977 (78%) completed the LEAP survey. Fig 2 shows recruitment and enrollment details. For the validation substudy, authorization for medical records access was requested from the 656 eligible LEAP survey respondents (those who reported at least one live birth); of these, 379 (58%) returned a signed HIPAA Authorization form and a medical record release form between July 2019 and January 2021. Study staff then sent the medical record release forms to the relevant providers, requesting prenatal, delivery, and postpartum medical records for each live birth reported by participants.

Medical records were abstracted for a total of 250 participants (66% of the 379 who consented; 38% of the 656 eligible participants). Of the 129 consenting participants whose records were not abstracted, records for 37 were not requested because the consents were received at the end of the data collection period; further, we had anticipated being able to fill in the pre-pregnancy and GWG data for these participants from birth record data requested from the Minnesota Department of Health. However, due to COVID-19 pressures, we never received the birth records data. A further ninety-two records were requested but no records were provided. Key reasons records were not provided were: (1) patient records could not be found by medical records staff; (2) medical records staff could identify only partial patient records; and/ or (3) the care facility name provided by the participant was incorrect and study staff were unable to obtain the correct information upon follow-up.

**Characteristics of the overall cohort and validation subsample.** Table 3 shows characteristics of the overall LEAP cohort and the LEAP validation subsample. Women in the overall cohort were largely White and had a high level of education. Cohort members were more likely to report White race, and less likely to report Black race than the Minnesota population (cohort: 66% White and 9% Black; Minnesota 2020 Census: 62% White and 12% Black). Mean adolescent BMI was at the 66th percentile relative to CDC growth standards [49], and 30% of the cohort had a BMI that met the CDC threshold for overweight or obesity ($\geq$85th percentile relative to standard for age and sex) [49]. Past-year prevalence of self-reported binge eating and clinically significant depressive symptoms in adolescence were 10% and 18%, respectively. One in three women reported a history of one or more types of child abuse (physical, sexual, or emotional) or neglect (physical or emotional). Most of the LEAP cohort (79%) had attempted a pregnancy, with two-thirds reporting a pregnancy that resulted in a live birth. Forty-two percent (n = 263) of parous women reported pre-pregnancy BMI $\geq$ 25 kg/m$^2$. On average, women in the cohort gained 37.2 pounds during their first pregnancy. Compared to the overall cohort, the validation subsample was more likely to be White (77% vs. 66%), and was slightly more likely to have a Bachelor's degree (58% vs. 55%).

**Table 3. Selected LEAP cohort and validation subsample characteristics.**

| | Overall Cohort (n = 977) | Validation Subsample* (n = 250) |
|---|---|---|
| **Variable** | **% (n) or Mean (SD)** | **% (n) or Mean (SD)** |
| Age at time of LEAP survey (years) | 36.2 (1.6) | 36.3 (1.5) |
| Race (EAT-I, age 11–18 years) | | |
| White | 66% (635) | 77% (189) |
| Black | 9% (84) | 6% (14) |
| Hispanic | 4% (37) | 0.8% (2) |
| Asian | 17% (159) | 10% (25) |
| Other | 5% (51) | 7% (16) |
| Childhood socioeconomic status based on parents' education (EAT-I, age 11–18 years) | | |
| Low | 13% (124) | 10% (25) |
| Low middle | 15% (143) | 14% (34) |
| Middle | 24% (234) | 26% (65) |
| Upper middle | 30% (284) | 34% (84) |
| Upper | 18% (174) | 17% (41) |
| Adult educational attainment (EAT-IV, age 25–36 years) | | |
| ≤High school | 19% (172) | 17% (40) |
| Some college or vocational training | 25% (226) | 26% (62) |
| ≥Bachelor's degree | 55% (495) | 58% (138) |
| Childhood risk factors and health indicators | | |
| BMI percentile in adolescence (EAT-I, age 11–18 years) | 65.8 (26.0) | 64.4 (25.7) |
| Binge eating in adolescence (EAT-I, age 11–18 years) | 10% (94) | 11% (27) |
| Unhealthy weight control behaviors (EAT-I, age 11–18 years) | 54% (518) | 56% (139) |
| Body dissatisfaction in adolescence (EAT-I, age 11–18 years) | 41% (394) | 41% (101) |
| Clinically significant depression symptoms in adolescence (KD≥23; EAT-I, age 11–18 years) | 18% (174) | 19% (47) |
| History of childhood abuse or neglect retrospectively reported in young adulthood (EAT-IV; age 25–36) | 37% (325) | 32% (76) |
| Reproductive and perinatal characteristics (LEAP) | | |
| Attempted pregnancy | 79% (769) | 100% (250) |
| Had at least one live birth | 67% (656) | 100% (250) |
| Maternal age among women with live births (years; first birth) | 27.1 (5.3) | 27.6 (5.0) |
| Gestation length among live births (weeks; first birth) | 39.1 (2.4) | 39.0 (2.4) |
| Multiple births among live births | 2% (11) | 2% (4) |
| Pre-pregnancy BMI among women with live births (kg/m$^2$; first birth) | 25.3 (5.9) | 25.8 (5.9) |
| Total GWG among women with live births (pounds; first birth) | 37.2 (27.2) | 35.6 (25.8) |
| GWG z-score among women with live births (SD; first birth) | -0.02 (SD = 1.2) | -0.12 (SD = 1.1) |
| High GWG z-score (>80th percentile; first birth) | 20% (120) | 18% (43) |
| Low GWG z-score (<20th percentile; first birth) | 18% (111) | 16% (39) |

*LEAP participants with at least one live birth from whom the study team received HIPAA and medical records release authorization and for whom the team was able to access their prenatal, delivery, and/or postpartum medical records.

Abbreviations: BMI, body mass index; EAT, Eating and Activity in Teens and Young Adults (number indicates survey wave: EAT-I is baseline at age 11–18 and EAT-IV is wave four at age 25–36); GWG, gestational weight gain

KD, Kandel and Davies Depressive Mood Scale score.[52]

Note: Low GWG is defined as a GWG z-score <20th percentile; moderate GWG 20th percentile ≤ z-score ≤ 80th percentile; and high GWG as z-score > 80th percentile. BMI percentiles in adolescence are relative to the standard for age and sex, based on the 2000 CDC growth charts.[49] Reproductive and perinatal characteristics are based on self-report on the LEAP survey. Observations summed across variable categories may differ from the total N due to missingness.

**Table 4. Overall cohort characteristics by GWG category among women with at least one live birth.**

| Variable | Low GWG (n = 111) | Moderate GWG (n = 373) | High GWG (n = 120) |
| --- | --- | --- | --- |
| | Col % (n) or Mean (SD) | Col % (n) or Mean (SD) | Col % (n) or Mean (SD) |
| Age at time of LEAP (years) | 36.2 (1.5) | 36.5 (1.4) | 36.4 (1.6) |
| Race (EAT-I, age 11–18 years) | | | |
| White | 52% (57) | 72% (266) | 59% (68) |
| Black | 8% (9) | 7% (26) | 11% (13) |
| Hispanic | 3% (3) | 5% (18) | 3% (4) |
| Asian | 35% (39) | 11% (41) | 16% (18) |
| Other | 2% (2) | 5% (18) | 11% (13) |
| Childhood socioeconomic status based on parents' education (EAT-I, age 11–18 years) | | | |
| Low | 21% (23) | 12% (45) | 15% (17) |
| Low middle | 12% (13) | 17% (61) | 15% (17) |
| Middle | 30% (32) | 21% (78) | 32% (38) |
| Upper middle | 23% (25) | 33% (120) | 30% (35) |
| Upper | 14% (15) | 18% (65) | 9% (10) |
| Adult educational attainment (EAT-IV, age 25–36 years) | | | |
| ≤High school | 19% (19) | 15% (51) | 30% (33) |
| Some college or vocational training | 22% (22) | 24% (83) | 35% (38) |
| ≥Bachelor's degree | 60% (61) | 61% (211) | 35% (38) |
| Childhood risk factors and health indicators | | | |
| BMI percentile in adolescence (EAT-I, age 11–18 years) | 58.4 (25.6) | 64.8 (25.7) | 67.2 (25.2) |
| Binge eating in adolescence (EAT-I, age 11–18 years) | 8% (9) | 11% (38) | 11% (12) |
| Unhealthy weight control behaviors (EAT-I, age 11–18 years) | 48% (53) | 55% (203) | 65% (77) |
| Body dissatisfaction in adolescence (EAT-I, age 11–18 years) | 38% (41) | 39% (143) | 44% (52) |
| Clinically significant depression symptoms in adolescence (KD≥23; EAT-I, age 11–18 years) | 15% (16) | 20% (75) | 15% (18) |
| History of childhood abuse or neglect retrospectively reported in young adulthood (EAT-IV; age 25–36) | 38% (38) | 32% (109) | 41% (44) |
| Perinatal characteristics, first birth | | | |
| Maternal age (years) | 26.8 (5.7) | 28.1 (4.9) | 25.6 (5.5) |
| Multiple births | 1% (1) | 1% (5) | 3% (3) |
| Gestation length (weeks) | 38.7 (2.5) | 39.3 (2.3) | 39.0 (2.7) |
| Pre-pregnancy BMI (kg/m$^2$) | 23.5 (3.8) | 25.8 (5.9) | 25.3 (6.2) |
| GWG (pounds) | 16.8 (4.6) | 31.1 (8.1) | 63.8 (22.7) |

**Characteristics of those with high, moderate, and low GWG z-scores.** Women categorized as having high GWG for their pre-pregnancy BMI and gestation length were less likely to have a bachelor's degree or higher education than women with moderate or low GWG (Table 4). Women with high GWG also differed from those with moderate or low GWG on certain adolescent characteristics. Women with high GWG had a higher adolescent BMI percentile (67.2) than those with moderate (64.8) or low (58.4) GWG. Women with high GWG were also substantially more likely to have engaged in unhealthy weight control behaviors in adolescence (65%) than those with moderate (55%) or low (48%) GWG. Lastly, women with high GWG had a greater prevalence of childhood maltreatment and adolescent body dissatisfaction than those with moderate GWG.

**Ongoing analyses.** Analyses are underway to validate survey-based pre-pregnancy weight and GWG measures against medical records data. These analyses will assess the degree of

misclassification of pre-pregnancy BMI and high and low GWG in survey reports, and examine patterns of misclassification across participant characteristics. Other analyses are being conducted to identify childhood, adolescent, and young adult predictors of high pre-pregnancy BMI and GWG, using the validation data to adjust outcomes for errors in survey reports. These analyses build on the descriptive results presented here, which suggest that certain childhood and adolescent characteristics (e.g., unhealthy weight control behaviors, body dissatisfaction, childhood maltreatment) may be associated with high or low GWG after adjustment for pre-pregnancy BMI and gestation length. Although the small number of multiple births in the cohort are included in the present analyses to provide a comprehensive overview of the cohort data, planned analyses will exclude them due to the differences in expected GWG in multiple versus singleton pregnancies. Analyses will use a multiple-imputation approach to adjust survey data on pregnancy outcomes for deviations from medical record data and include rigorous control for confounding variables. Funding is currently being sought for follow up of the cohort, which will allow for analyses of the ways that pregnancy weight and behaviors affect long-term outcomes for women.

## Strengths and limitations

*Strengths*. The key strengths of LEAP are the linkage of prospective data collected across the life course with data on pregnancy. Because of the focus of Project EAT, the cohort from which we recruited the LEAP sample, LEAP is particularly well-suited to investigating weight-related health and behaviors. The LEAP survey included a wide range of questions on social, behavioral, and physical factors in pregnancy that are potentially relevant to pregnancy-related health, allowing for investigation of a diverse set of exposures and outcomes. Lastly, medical record data from the validation subsample allow for investigation of the accuracy of key pregnancy characteristics. This validation substudy provides important information on the likely degree of bias in other studies using self-report of these outcomes and allows us to conduct bias-adjusted analyses in the LEAP cohort study. The validation data on pregnancy weight measures distinguishes LEAP from studies that have also retrospectively collected pregnancy data such as the NLSY.

*Limitations*. Several limitations of this study should be noted. First, we relied on retrospectively-collected pregnancy data. We chose this approach because it is generally not feasible to follow pregnancies prospectively within a longitudinal cohort, due to the small number of women who give birth in any given year. Although we are able to validate survey reports of key pregnancy characteristics (e.g., GWG, pregnancy complications) using medical records, other potentially important dimensions of participants' pregnancies, such as social factors and health behaviors (financial stress, disordered eating), which were also retrospectively reported, cannot be validated because there is no prospectively measured source of data against which they can be compared. A second limitation is attrition over time in the prospective cohort from which LEAP participants were recruited, which may cause selection bias. Third, the validation substudy comprises a minority (38%) of the survey cohort and this subsample may differ from the overall cohort. Fourth, the LEAP cohort is disproportionately White and well-educated, limiting our capacity to answer important health equity questions. Finally, the sample size limits our ability to look at low-frequency but important pregnancy outcomes, such as pre-eclampsia.

## Conclusion

High pre-pregnancy weight and excessive GWG are thought to have important effects on the health of women and their children. Very few risk factors for excess GWG have been

identified. Growing evidence suggests that many health risks develop over the life course, and that risk factors and behaviors in childhood and adolescence can have long-term consequences for adult health. The linkage of longitudinal, prospectively-collected weight-related data to retrospectively-reported pregnancy information validated against medical records provides an important foundation for understanding life course influences on weight-related health and behavior during pregnancy.

## Supporting information

**S1 File. LEAP survey questions.**
(DOCX)

## Author Contributions

**Conceptualization:** Susan M. Mason, Lisa M. Bodnar, Dianne Neumark-Sztainer.

**Data curation:** Kriszta Farkas, Jessica K. Friedman, Anne Gerlach, Sydney T. Johnson, Rebecca Emery Tavernier.

**Formal analysis:** Kriszta Farkas.

**Funding acquisition:** Susan M. Mason.

**Methodology:** Susan M. Mason, Lisa M. Bodnar.

**Project administration:** Susan M. Mason, Anne Gerlach, Sydney T. Johnson, Rebecca Emery Tavernier.

**Resources:** Dianne Neumark-Sztainer.

**Supervision:** Susan M. Mason, Anne Gerlach.

**Validation:** Jessica K. Friedman.

**Writing – original draft:** Susan M. Mason.

**Writing – review & editing:** Kriszta Farkas, Jessica K. Friedman, Anne Gerlach, Sydney T. Johnson, Rebecca Emery Tavernier, Lisa M. Bodnar, Dianne Neumark-Sztainer.

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
