## [Decision Letter · Decision Letter 0]

28 Jul 2023

PONE-D-23-18936Cohort Profile: Life-course Experiences And Pregnancy (LEAP) – A retrospective pregnancy cohort for life-course reproductive health researchPLOS ONE

Dear Dr. Mason,

Thank you for submitting your manuscript to PLOS ONE. After careful consideration, we feel that it has merit but does not fully meet PLOS ONE’s publication criteria as it currently stands. Therefore, we invite you to submit a revised version of the manuscript that addresses the points raised during the review process.

We look forward to receiving your revised manuscript.

Kind regards,

Andrea Maugeri

Academic Editor

PLOS ONE

Journal Requirements:

Reviewers' comments:

Reviewer's Responses to Questions

**Comments to the Author**

1. Is the manuscript technically sound, and do the data support the conclusions?

Reviewer #1: Yes

Reviewer #2: Partly

2. Has the statistical analysis been performed appropriately and rigorously? 

Reviewer #1: Yes

Reviewer #2: No

3. Have the authors made all data underlying the findings in their manuscript fully available?

Reviewer #1: Yes

Reviewer #2: No

4. Is the manuscript presented in an intelligible fashion and written in standard English?

Reviewer #1: Yes

Reviewer #2: Yes

5. Review Comments to the Author

Reviewer #1: The authors provided initial, descriptive results from the Life-course Experiences And Pregnancy (LEAP) study. The authors described the cohort well. I have few comments:

1. A detailed plan of analysis is needed.

2. A conceptual framework will help the readers to understand the objectives of the cohort.

3. How the findings from the cohort will help to fill the knowledge gap in this domain?

Reviewer #2: Paper “Cohort Profile: Life-course Experiences And Pregnancy (LEAP) – A retrospective

pregnancy cohort for life-course reproductive health research” is a description of the study population and methods used in the Life Course and Pregnancy (LEAP) study, which is nested in Eating and Activity in Teens and Young Adults (EAT) study. The authors' aim was to provide an extensive description of study design, population, recruitment, data gathering, and validation. Additionally, authors provide information about pre-pregnancy characteristics in those whose GWG was within, above, or below IOM recommendations.

The study is retrospective in design, but the authors provided a good rationale for conducting the retrospective study. The strengths of the study are the methods used, with care taken on data validity. The paper is well-written and easy to understand.

Although very good overall impression I have two major and few minor remarks. In my opinion, the paper would benefit if authors consider the problems described below:

Major issue:

1) Calculation of GWG should be corrected taking pregnancy week into account, for example as described here: https://onlinelibrary.wiley.com/doi/10.1002/oby.20951

This correction could potentially change the results and conclusions of the study. You should also check if other sections need re-writing after the adjustment.

2) When providing data regarding the population and its subgroups, use statistical methods to show if there are statistically significant differences, and report them along with p-value.

Minor issues:

3) Did you analyze twins separately? GWG in twins is different from singleton and this should be taken into account.

4) A low response rate is a serious limitation, and it should be mentioned. Only 38% of those eligible were included in the study.

5) Why did exclude those living outside of the US? How many were excluded?

6) You have mentioned White overrepresentation, what are race proportions in your region? What are the expected racial proportions in the study?

7) In the cohort description, you mention 2$ incentive. This is a much different amount from the other incentives you mentioned. Is this correct?

6. PLOS authors have the option to publish the peer review history of their article (what does this mean?). If published, this will include your full peer review and any attached files.

Reviewer #1: **Yes: **Ranadip Chowdhury

Reviewer #2: **Yes: **Jan Modzelewski

---

## [Author Response · Author response to Decision Letter 0]

28 Sep 2023

Editor comments:

1. Please ensure that your manuscript meets PLOS ONE's style requirements. 

Response: We have updated the formatting to comply with PLOS ONE’s style requirements.

Response: We have removed funding acknowledgements from the manuscript.

3. Upon re-submitting your revised manuscript, please upload your study’s minimal underlying data set as either Supporting Information files or to a stable, public repository and include the relevant URLs, DOIs, or accession numbers within your revised cover letter. We will update your Data Availability statement to reflect the information you provide in your cover letter.

Response: We have uploaded the minimal underlying dataset to OpenICPSR. The data can be accessed at http://doi.org/10.3886/E193505V1.

Response: The data are now available at OpenICPSR at http://doi.org/10.3886/E193505V1.

Response: Detail on informed consent for the survey is now provided on page 10 and for the medical records on page 11. The full name of the IRB is included in the manuscript on page 1. 

Response: Captions for Supporting Information files have been added at the end of the manuscript. 

Reviewers' comments:

Comments to the Author

Reviewer #1: The authors provided initial, descriptive results from the Life-course Experiences And Pregnancy (LEAP) study. The authors described the cohort well. I have few comments:

1. A detailed plan of analysis is needed.

Response: 

We have now added additional details on our analysis plans on pages 20-22. In addition to adding more detail on variables included in analyses, we have included the following description of the analyses conducted: 

“Preliminary analyses were descriptive and designed give an overview of the key characteristics of the LEAP cohort overall and the medical record validation subsample. In addition, we present characteristics of those with low, moderate, and high GWG z-scores. 

First, we calculated the distribution (number and percent for categorical and dichotomous variables, means and SDs for continuous variables) of women in the overall sample across the sociodemographic, childhood and adolescent risk factor and health indicator, and reproductive perinatal variables described above. To allow for exploration of the extent to which the validation subsample differed from the overall cohort, we also present the distribution of the subsample across the same characteristics. Finally, because a primary outcome of interest in the LEAP study is GWG, we examine the distribution of these characteristics across categories of GWG. 

We did no hypothesis testing, given growing consensus about the limited information it provides as well as the harms of its widespread use (47,48), and instead focus on descriptive statistics and qualitative comparison across GWG groups.”

2. A conceptual framework will help the readers to understand the objectives of the cohort.

Response: 

We have added a conceptual model (Figure 1) illustrating the core hypotheses and the aims of LEAP, and we have described this conceptual model in the Introduction (page 10) as follows:

“The specific focus of LEAP is to identify the extent to which childhood maltreatment and its sequelae, including adolescent and young adult disordered eating and depression, are linked to high pre-pregnancy BMI and excessive GWG. The conceptual model guiding the LEAP study is shown in Figure 1, which illustrates the theory that child maltreatment victimization reduces the capacity for affect regulation (the ability to cope with distress), leading to obesity-promoting behaviors that are soothing or numbing, such as overeating of highly palatable foods that trigger dopaminergic reward responses (32–37). Associations between child maltreatment, disordered eating, and body weight have been repeatedly demonstrated outside of pregnancy.”

3. How the findings from the cohort will help to fill the knowledge gap in this domain?

Response:

We have now provided additional information on the knowledge gaps and how this cohort provides appropriate data infrastructure to address these gaps. We have added the following text on page 10:

“The LEAP cohort was designed to support research on life course determinants of pregnancy weight-related health, and to fill a major gap in existing data linking adolescent weight-related risk factors to validated pregnancy outcomes. This linkage across the life course will allow us to assess whether certain environmental, psychological, or behavioral factors in adolescence predict weight-related markers in pregnancy, informing an understanding of the life course etiology of pregnancy weight-related health.”

 

Reviewer #2: Paper “Cohort Profile: Life-course Experiences And Pregnancy (LEAP) – A retrospective pregnancy cohort for life-course reproductive health research” is a description of the study population and methods used in the Life Course and Pregnancy (LEAP) study, which is nested in Eating and Activity in Teens and Young Adults (EAT) study. The authors' aim was to provide an extensive description of study design, population, recruitment, data gathering, and validation. Additionally, authors provide information about pre-pregnancy characteristics in those whose GWG was within, above, or below IOM recommendations. The study is retrospective in design, but the authors provided a good rationale for conducting the retrospective study. The strengths of the study are the methods used, with care taken on data validity. The paper is well-written and easy to understand.

Although very good overall impression I have two major and few minor remarks. In my opinion, the paper would benefit if authors consider the problems described below:

Major issue:

1) Calculation of GWG should be corrected taking pregnancy week into account, for example as described here: https://onlinelibrary.wiley.com/doi/10.1002/oby.20951

This correction could potentially change the results and conclusions of the study. You should also check if other sections need re-writing after the adjustment.

Response:

To address this comment, we re-ran analyses with the gestational weight gain (GWG) z-score rather than using IOM categories. The GWG z-score is calculated based on distribution of GWG at each gestation week, separately for each pre-pregnancy BMI category. Thus, it corrects for differences in GWG that arise from different gestation lengths. This measure is described in the Methods on page 14. Results are largely the same as with the IOM categories. The updated results can be seen in the GWG variables in Table 2a, throughout Table 2b, and in the results text. 

2) When providing data regarding the population and its subgroups, use statistical methods to show if there are statistically significant differences, and report them along with p-value.

Response:

Although we realize some readers will expect p-values, we do not conduct null-hypothesis significance testing (p-values) in our manuscript based on recommendations from the American Statistical Association’s Statement on Statistical Significance and P-Values and other sources. To address this comment, we have now stated our rationale for avoiding p-values in the manuscript and have provided the following references that have informed our position (page 22):

47. Amrhein V, Greenland S, McShane B. Scientists rise up against statistical significance. Nature. 2019 Mar;567(7748):305–7. 

48. Wasserstein RL, Lazar NA. The ASA Statement on p-Values: Context, Process, and Purpose. The American Statistician. 2016 Apr 2;70(2):129–33. 

Minor issues:

3) Did you analyze twins separately? GWG in twins is different from singleton and this should be taken into account.

Response:

Twins were included in the analyses presented here, as we wished to comprehensively present the data on our cohort. We have now added information on the number of twin births (n=10) included in Table 2a. We have also noted (page 28) that in upcoming planned analyses twins will be excluded due to the different expected GWG in twin vs. singleton pregnancies.

4) A low response rate is a serious limitation, and it should be mentioned. Only 38% of those eligible were included in the study.

Response:

There are two issues of concern with regard to response: attrition from the original cohort, and response rate to the current survey and medical records validation. Attrition from the original cohort was substantial over 20 years of follow-up, in part because the parent study (Project EAT) was not originally designed to follow participants and thus contact information for follow-up was not comprehensive. The drop-off in follow-up is shown in Figure 2. Within the cohort still being followed, the response rate to the LEAP survey, the main source of data for this cohort, was relatively high (78%). For the validation study, the response rate to the request for medical records was 58% (379 consents / 656 invitations). We did not expect to achieve a higher response rate, as many participants feel uncomfortable providing their medical records. The final number of participants included in the validation study was 250 (38% of the 656 invited); the reduction from 379 to 250 was due to the inability to track down medical records for 129 participants. Notably, the women in the validation subsample are not substantially different from the larger LEAP cohort on most measured characteristics. Of the issues described above, we believe attrition from the main cohort is likely the largest potential source of bias. We have described this in the limitations and have also added concerns about the representativeness of the validation substudy given that it represents only a subset of the sample (page 19). Figure 2 provides a flow chart of sample exclusions and inclusions. 

5) Why did exclude those living outside of the US? How many were excluded?

Response:

Four women without a US-based mailing address were excluded from the 1252 invited to participate in the LEAP survey. Given the cost and complexity of recruiting internationally, we chose to focus on women residing in the US. The cohort flow chart (Figure 2) now includes this detail.

6) You have mentioned White overrepresentation, what are race proportions in your region? What are the expected racial proportions in the study?

Response:

We have now provided additional detail on the population proportions of these groups for comparison against our sample.

We have added the following text on page 23:

“Cohort members were more likely to report White race and less likely to report Black race than the Minnesota population (cohort: 66% White and 9% Black; Minnesota 2020 Census: 62% White and 12% Black).”

7) In the cohort description, you mention 2$ incentive. This is a much different amount from the other incentives you mentioned. Is this correct?

Response:

Thank you for this question. Our description of the $2 bill was incorrect. It was not an incentive because we did not provide it in response to survey completion. Rather, we provided the $2 bill in the final recruitment mailings to non-responders to increase attention to the mailing. We have now clarified this in the manuscript (page 12).

---

## [Editor Report · Decision Letter 1]

30 Nov 2023

Cohort Profile: Life-course Experiences And Pregnancy (LEAP) – A retrospective pregnancy cohort for life-course reproductive health research

PONE-D-23-18936R1

Dear Dr. Mason,

We’re pleased to inform you that your manuscript has been judged scientifically suitable for publication and will be formally accepted for publication once it meets all outstanding technical requirements.

Kind regards,

Andrea Maugeri

Academic Editor

PLOS ONE
---

## [Editor Report · Acceptance letter]

22 Feb 2024

PONE-D-23-18936R1 

PLOS ONE

Dear Dr. Mason, 

I'm pleased to inform you that your manuscript has been deemed suitable for publication in PLOS ONE. Congratulations! Your manuscript is now being handed over to our production team.

Kind regards, 

on behalf of

Dr. Andrea Maugeri 

Academic Editor

PLOS ONE